# Automatic Power Optimization of a 44 Tbit/s Real-Time Transmission System over 1900 km G.654.E Fiber and Widened C+L Erbium-Doped Fiber Amplifiers Utilizing 400 Gbit/s Transponders

Anxu Zhang [1], Yuyang Liu [1], Lipeng Feng [1,*], Huan Chen [2], Yuting Du [2], Jun Wu [3], Kai Lv [1], Hao Liu [1], Xia Sheng [1] and Xiaoli Huo [1]

[1] State Key Laboratory of Optical Fiber and Cable Manufacture Technology, China Telecom Research Institute, Beijing 100876, China; zhanganx@chinatelecom.cn (A.Z.); liuyy26@chinatelecom.cn (Y.L.); lvkai@chinatelecom.cn (K.L.); liuhao3@chinatelecom.cn (H.L.); shengx3@chinatelecom.cn (X.S.); huoxl@chinatelecom.cn (X.H.)

[2] WDM System Design Department of Wireline Product R&D Institute, ZTE Corporation, Wuhan 430073, China; chen.huan6@zte.com.cn (H.C.); du.yuting@zte.com.cn (Y.D.)

[3] State Key Laboratory of Optical Fiber and Cable Manufacture Technology, Yangtze Optical Fiber and Cable Joint Stock Limited Company, Wuhan 430073, China; wujun_02737@yofc.com

[*] Correspondence: fenglp@chinatelecom.cn

**Abstract:** Power unevenness, mainly induced by stimulated Raman scattering, has been a major problem in multi-band transmission systems, especially in the upcoming field-deployed 400 Gbit/s widened C band plus L band system for backbone long-haul and ultra-long-haul scenarios. To reduce the impact of power unevenness, we propose an automatic power optimization (APO) algorithm to guarantee reliable transmission for all channels, especially the channels at short wavelengths. The simulation results show that the power unevenness of output power after 5-span transmission in the C band is up to 11 dB before APO, while after APO is applied, it is greatly improved to less than 1.6 dB. To further investigate the performance of the APO scheme, we conduct a real-time 44 Tbit/s C+L transmission system over 1900 km G.654.E fiber utilizing 400 Gbit/s transponders. The experimental results show that the power unevenness has been effectively compensated from 12 dB to 4 dB in the entire 11 THz range, making the received optical signal-to-noise ratio relatively flat (3.4 dB). Moreover, the capacity and distance product of this system is 83.6 Pbit/s·km (44 Tbit/s × 1900 km), and to the best of our knowledge, this is a record capacity and distance product in the real-time single-mode fiber transmission system.

**Keywords:** real-time 400 Gbit/s transmission; automatic power optimization; G.654.E fiber; widened C+L band

## 1. Introduction

With the rapid development of bandwidth-intensive services such as 5G, cloud computing, mobile internet, and ultra-high-definition video, network traffic will continue to maintain a rapid growth. According to Omdia, from 2017 to 2024, the compound annual growth rate (CAGR) of cellular networks and consumer-fixed broadband networks was as high as 28.7% [1]. The ever-increasing demand for data transmission capacity has driven the widespread deployment of the coherent 100 Gbit/s and 200 Gbit/s dense wavelength-division-multiplexing (DWDM) systems in the long-haul (LH) and ultra-long-haul (ULH) backbone optical networks around the world [2–4]. In the upcoming few years, 400 Gbit/s DWDM systems with LH and ULH transmission capabilities are expected to be deployed on a large scale. Moreover, the capacity demand has also inspired the investigation of new-type fibers with spatial dimensions and additional spectral windows [5–8]. Compared

with the spatial-division-multiplexing (SDM) fiber system, adopting the high data rate and multi-band transmission (MBT) windows should be a promising approach in the short term, because the construction period of new optical fibers usually takes 2–3 years. There have been substantial efforts to implement the MBT systems to the optical transmission network. In the lab test validation, recently, using discrete amplifiers covering a 97 nm wavelength range, a 63.2 Tbit/s system transmitting over 440 km has been demonstrated, which shows a feasible implementation of the MBT system in the metropolitan networks [9]. In [10], a 112.8 Tbit/s real-time transmission over 101 km G.654.E fiber has been demonstrated, which is amplified by thulium-doped fiber amplifiers (TDFA) for S band and erbium-doped fiber amplifiers (EDFA) for C+L band with the help of hybrid forward- and backward-pumped distributed Raman amplifiers. Moreover, field trials of high-speed the MBT have also been frequently demonstrated. For example, a record 158.4 Tbit/s transmission with offline digital signal processing (DSP) over 2x60 km field deployed single mode fiber (SMF) using S+C+L 18 THz bandwidth lumped amplification is demonstrated in [11], showing the feasibility of relying on the MBT to handle high-capacity demand. In [12], an O+S+C+L+U-band transmission system is demonstrated with a record bandwidth of 25 THz over a 45 km deployed fiber-optic cable. Moreover, the MBT system can also be observed in the co-fiber transmission scenarios of quantum key distribution (QKD) [13–16] and classical communication. Due to the low power and sensitivity to noise of the QKD signal, the QKD signal and the classical signal should be carefully allocated in different frequency bands to reduce the performance degradation caused by the coexistence signals. One of the main challenges of MBT systems is the performance degradation caused by cumulative power unevenness, especially in the case of large bandwidth, where the power transfer caused by the stimulated Raman scattering (SRS) effect dominates [17,18]. The specific impact of the SRS effect is that in the MBT system, it transfers the power of the signals from short wavelength to long wavelength. Such power transfer will deteriorate the optical signal-to-noise ratio (OSNR) performance and further limit the transmission distance and maximum achievable transmission capacity. Thus, overcoming SRS effect-induced power transfer is quite meaningful and essential for improving the transmission performance of optical communication systems, especially for the high-rate and large capacity LH and ULH systems.

In this paper, an automatic power optimization (APO) algorithm suitable for practical engineering applications has been proposed, which only needs the power spectrum information of the transmitter, the receiver, and the dynamic gain equalization (DGE) stations. We demonstrate the feasibility of the proposed algorithm by building a 5-span fiber system with 11 THz bandwidth in the simulation and experiments. Moreover, we extend our previous work in [19] and experimentally investigate a 44 Tbit/s real-time DWDM transmission over 1900 km G.654.E SMF link utilizing 400 Gbit/s transponders with a symbol rate of 91.6 GBaud. Commercially available C band and L band WSSs and EDFAs with 11 THz bandwidth are adopted to generate and amplify 110 DWDM channels spaced by 100 GHz. The optical back-to-back (B2B) performance is demonstrated and shows a required optical signal-to-noise ratio (OSNR) of ~17 dB. Received OSNR and long-term Q-factors are tested. The capacity and distance product of this system is 83.6 Pbit/s·km (44 Tbit/s × 1900 km), and to the best of our knowledge, this is a recorded capacity and distance product in the real-time SMF transmission system.

## 2. Principle

Power unevenness has been a major problem in the MBT system, where power transfer caused by the SRS effect has a significant impact on power unevenness. The SRS effect transfers the power of the signals from short wavelength to long wavelength. Moreover, the resulting power transfer will deteriorate the OSNR performance and further limit the transmission distance and maximum achievable transmission capacity.

The model of the SRS effect-induced power transfer is introduced as follows. At first, the power evolution of the $k$-th channel along the fiber, $S_k$, can be represented as:

$$\frac{dS_k}{dz} = -\alpha S_k + S_k \sum_{i=1}^{N} \frac{\gamma_{ik}}{2A_{eff}} S_i \tag{1}$$

where $\gamma_{ik}$ represents the Raman gain coefficient between the $i$-th channel and the $k$-th channel, $A_{eff}$ represents the effective area, and $\alpha$ is the fiber attenuation coefficient. Assuming that the Raman gain spectrum is a triangular gain spectrum, it can be calculated that for a certain length of fiber, the power transmitted to different wavelengths at distance $z$ can be expressed as:

$$S(z,\lambda) = \frac{S(0,\lambda)P_0 e^{-\alpha z}}{\int S(0,\Lambda) e^{[\beta P_0 L_{eff}(\Lambda-\lambda)]} d\lambda} \tag{2}$$

where $P_0$ represents the total optical input power and $S(0,\lambda)$ represents the power input from the optical fiber to each channel. Moreover, the effective length $L_{eff}$ and Raman transfer coefficient $\beta$ can be calculated as follows:

$$L_{eff} = \frac{1 - e^{-\alpha z}}{\alpha} \tag{3}$$

$$\beta = \frac{\gamma_{ik}}{2A_{eff}(\lambda_k - \lambda_i)} \tag{4}$$

To evaluate the impact of power unevenness and demonstrate the APO strategies, a simulated transmission system with the C band (spectrum bandwidth of 6 THz, C6T) and L band (spectrum bandwidth of 5 THz, L5T) is performed, and the physical layout is depicted in Figure 1. The fiber link is assumed to be composed of equal spans of 100 km with ITU-T G.654.E fiber. The fiber's attenuation curve and Raman gain curves are imported into the simulation system. The signals are amplified by two separated EDFAs through a C/L device before and after the amplification, and the insertion loss of the C/L device is set as 0.6 dB and 0.8 dB for the C and L bands, respectively. The span losses are adjusted to 22 dB and 22.5 dB for the C band and L band by simulating a variable optical attenuator (VOA). An ideal optical amplifier model is adopted for optical amplifiers (OAs), and the gain $G(\lambda)$ at the frequency of $\lambda$ can be expressed as:

$$G(\lambda) = G_0 + GainTilt \cdot (f_\lambda - f_{ref}) \tag{5}$$

where $G_0$ represents the average gain of the OA, $GainTilt$ is the gain slope, and $f_{ref}$ is the reference frequency. Here, the $G_0$ parameters at the C band and L band are, respectively, set as 22 dB and 22.5 dB to compensate for span losses, while the $GainTilt$ parameters of the C band and L band are, respectively, set as $-1.5$ dB and $-1$ dB, which are the same as the default values of the OAs in the experiment. The $f_{ref}$ is set as the center frequency of each band. The entire setup is defined as one optical multiplex section (OMS).

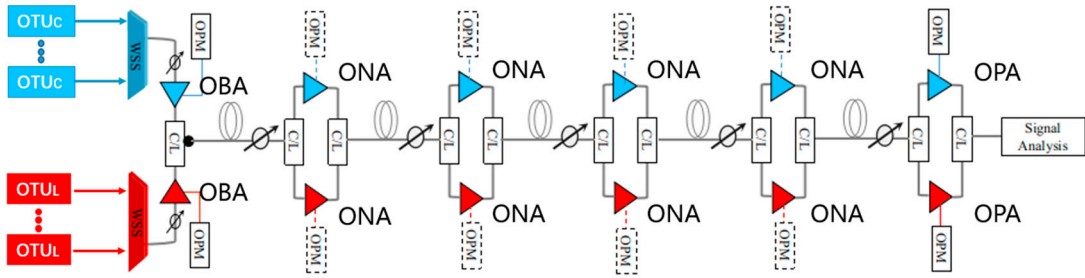

**Figure 1.** Simulation setup of the C6T and L5T transmission system. $OTU_{C/L}$: optical transform unit, C/L: C and L wavelength combiner, OBA: optical booster amplifier, ONA: optical node amplifier, OPA: optical pre-amplifier, OPM: optical power monitor, WSS: wavelength selective switch.

Figure 2 shows the power spectrum result at the receiver side after 5-span G.654.E fiber transmission. The optical input power per wavelength of the C band and L band is set to 4.8 dBm and 4 dBm, respectively. Shown as the red lines of Figure 2, the power flatness of the C band and L band at the receiver side, the difference between the maximum and minimum power values of all channels, is about 11 dB and 1.2 dB, respectively. This serious power unevenness will lead to a rapid OSNR degradation for some wavelengths, especially for short wavelengths in the C band, as shown in the red lines of Figure 2. Moreover, we also evaluate the performance of the system in terms of the OSNR and Q-factor, as Figure 3 shows. The simulation results indicate that after 5-span transmission, the OSNR of different channels range from 20.3 dB to 26.4 dB and from 25.3 dB to 27 dB for the C and L bands, respectively, and the worst Q-factor has also decreased to 6.1 dB.

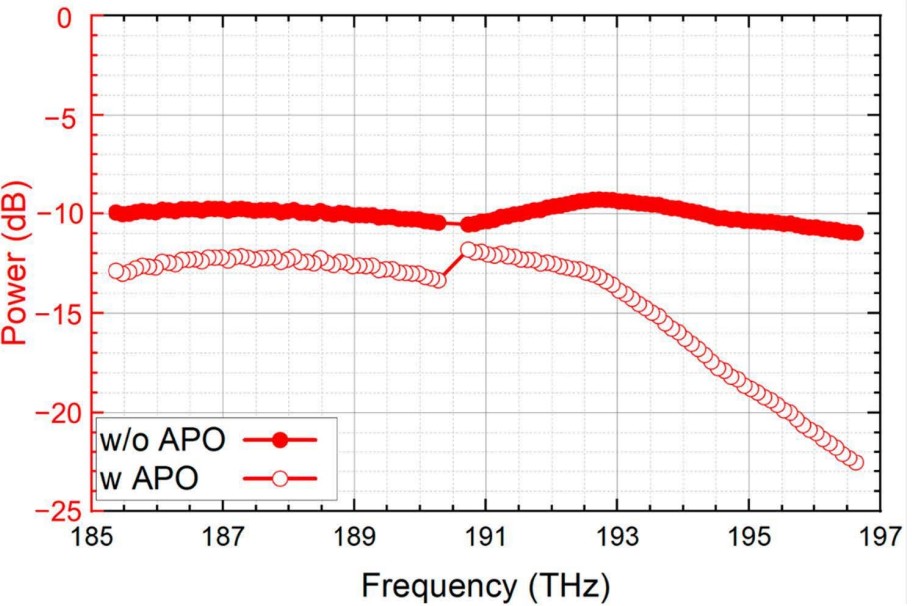

**Figure 2.** Power spectrum at the receiver side after 5-span G.654.E fiber transmission with and without the APO scheme.

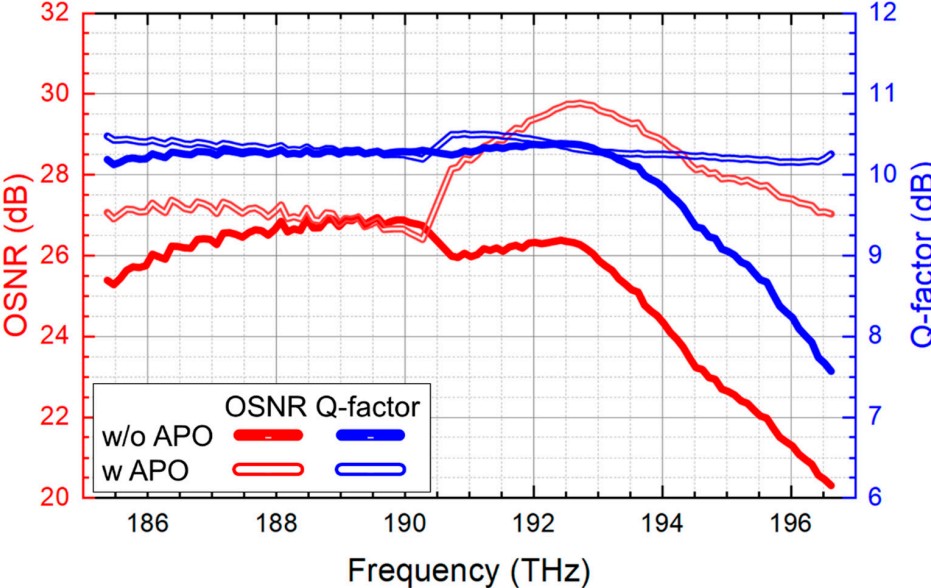

**Figure 3.** Performance comparison of the system with and without the APO algorithm in terms of the OSNR and Q-factor.

The procedure of the APO scheme is shown in Figure 4. In the initialization stage, we first adjust the gains of the C band and L band OAs to ensure that the output powers of the optical node amplifier (ONA) and the optical pre-amplifier (OPA) are the same as that of the optical booster amplifier (OBA).

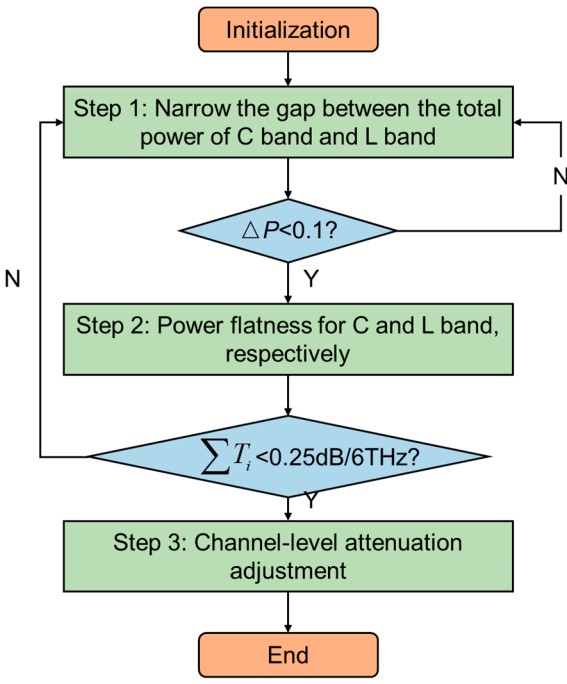

**Figure 4.** The procedure of the APO scheme.

Then, the first step (Step 1) is to narrow the power gap between the C band and L band by adjusting the gain of the C band and L band optical amplifiers. Specifically, it means increasing the output power of the C-band amplifiers and decreasing the output power of the L-band amplifiers. The increased/decreased power for all the C band and L band OAs can be represented as follows:

$$P_{out-diff} = \frac{\sum \Delta P}{2N} = \frac{\sum \left( P_{OA-L_{in}} - P_{OA-C_{in}} \right)}{2N} \tag{6}$$

where $\Delta P$ is the received optical power difference between the C band and L band of every span, and $N$ is the number of fiber spans. Step 1 needs to be iterated until $\Delta P < 0.1$, as the SRS-induced power transfer will affect the adjusted results.

Next, the second step (Step 2) is to separately compensate for power unevenness within the C band and L band by adjusting the slopes of the OAs. In practical engineering applications, the system only has an optical power monitor (OPM) at every span. The total slope compensation can be detected based on the OPM and then allocate the slope compensation to OAs in the link in a certain proportion. Assuming that the spectrum of the input signal is flat and the slope of the *m*-th fiber span caused by the power transfer is $T_m$ (dB/THz), to compensate for the power transfer, the slope compensation amount for each OA can be expressed as follows:

$$OA_{slope}(i) = \begin{cases} P \cdot T_1 & i = 1 \\ (1-P)T_1 + P \cdot T_2 & i = 2 \\ (1-P)T_2 + P \cdot T_3 & i = 3 \\ \quad \vdots & \quad \vdots \\ (1-P)T_{N-1} + P \cdot T_N & i = N \\ (1-P)T_N & i = N+1 \end{cases} \tag{7}$$

where $P$ is the adjustment ratio, and $N$ is the span number. When $P = 0$, the slope of the first OA is not adjusted, i.e., the power at the transmitter is still flat, which is called transmitter leveling. Similarly, when $P = 0.5$, this is called the transmitter-side half compensation (THC) strategy.

However, it should be noted that considering cost, OPM is currently only available at the start and end of OMS, that is, the OPM with solid line connections in Figure 1 can be used. This means that only the power spectrum at the first and last end of this OMS can be detected. The total slope $\sum\limits_{i=1}^{N} T_i$ is distributed to each OA. The proportion of allocation can be expressed as follows:

$$
OA_{slope}(i) =
\begin{cases}
P \dfrac{\sum\limits_{i=1}^{N} T_i}{N+1} & i = 1 \\[3mm]
\dfrac{\sum\limits_{i=1}^{N} T_i}{N+1} & i = 2 \\[3mm]
\dfrac{\sum\limits_{i=1}^{N} T_i}{N+1} & i = 3 \\[3mm]
\vdots & \vdots \\[3mm]
\dfrac{\sum\limits_{i=1}^{N} T_i}{N+1} & i = N \\[3mm]
(1-P) \dfrac{\sum\limits_{i=1}^{N} T_i}{N+1} & i = N+1
\end{cases}
\tag{8}
$$

It should be noted that after adjusting the slopes of OAs, the SRS effect in the system will change, so Step 1 and Step 2 need to be iterated repeatedly until they stabilize. The iteration conditions are shown in Figure 4.

Finally, the third step (Step 3) is to compensate for residual power unevenness by adjusting the channel attenuation through the C band and L band WSSs. The adjustment amount can be calculated as follows:

$$
WSS_{adj} = \frac{P_2 - P_0}{2}
\tag{9}
$$

where $P_2$ is the target power of the receiver side and $P_0$ is the actual received power.

Based on the consideration of both nonlinearity and OSNR performance, the power unevenness compensation is adjusted according to the THC strategy. As shown in the green line of Figure 3, it can be observed that after APO-based power compensation, the power unevenness in the C band is about 1.6 dB, and the power unevenness in the L band is about 1.2 dB, which means that the power unevenness can be effectively compensated. Moreover, by using the APO-based power compensation, the Q-factor performance of the system tends to be stable, which also indicates that the APO-based power compensation has an excellent power equalization ability.

## 3. Experimental Setup

The experimental setup is shown in Figure 5. At the transmitter side, 110 DWDM channels spaced by 100 GHz consist of 60 C band and 50 L band wavelengths. In the C band, one channel is filled with a 400 Gbit/s real-time signal with a symbol rate of 91.6 GBaud. The 400 Gbit/s real-time signal is realized with probabilistic constellation shaping (PCS) and polarization-division-multiplexed 16-point quadrature-amplitude modulation (PDM-16QAM). The remaining 59 channels are loaded with amplified spontaneous emission (ASE) noise, which is spectrally shaped by the concatenation of two EDFAs. The 400 Gbit/s signal and the 59 loading channels are multiplexed using a 6 THz widened C band WSS and then amplified using an EDFA. In the L band, similar to that of the C band configuration, the

50 channels consist of one 400 Gbit/s real-time signal and 49 loading channels. Thus, the total capacity is 44 Tbit/s (400 Gbit/s × 110 = 44 Tbit/s). The WSSs and EDFAs in the C band can support 48 nm optical bandwidth, ranging from 1524.4 nm to 1572.3 nm, and those of the L band can support 43.2 nm optical bandwidth, ranging from 1575.2 nm to 1618.4 nm. The C band multiplexed signals and the L band multiplexed signals are coupled and then sent to the fiber transmission link.

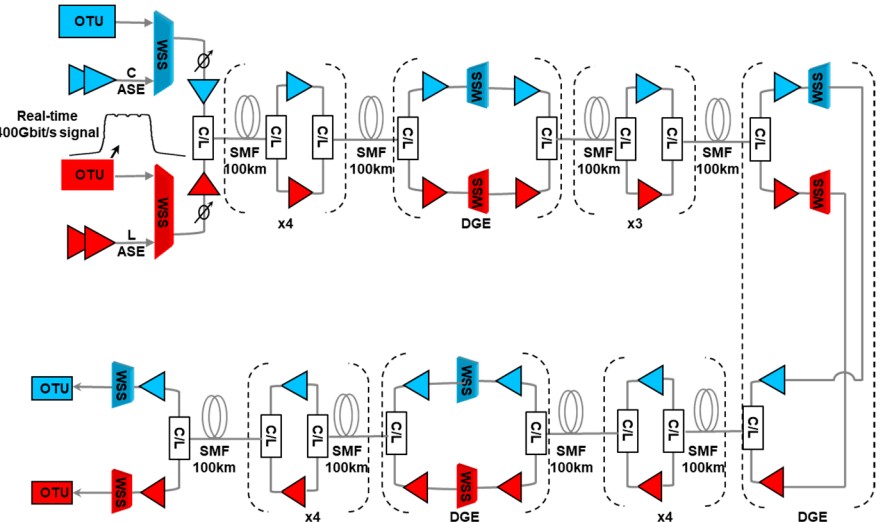

**Figure 5.** Real-time 400 Gbit/s transmission system setup using G.654.E fiber.

The G.654.E transmission link is composed of 19 spans with the adoption of 100 km SMF, EDFAs, and three DGE units. DGE consists of cascaded EDFAs, WSSes, and C/L devices in the C and L bands. The average loss of each span is 22 dB for the C band and 22.5 dB for the L band by adjusting the variable optical attenuator (VOA) inserted into each span. Span 1 to Span 4, Span 6 to Span 8, Span 10 to Span 13, and Span 15 to Span 18 are composed of 100 km SMF and one EDFA, whereas Span 5, Span 9, and Span 14 are composed of 100 km SMF and a DGE unit to adjust the optical power per wavelength. Meanwhile, the setup includes four OMSs, namely, from Span 1 to Span 5, Span 6 to Span 10, Span 11 to Span 15, and Span 16 to Span 19. Here, we adopt the THC strategy to compensate for the power unevenness. The whole 11-THz signal at the transmitter side is pre-tilted. While transmitting through the 1st OMS, the power of different wavelengths is affected differently by the SRS-induced power transfer, the optical amplifier (OA) slopes, and the wavelength-dependent loss of fiber. Meanwhile, the power unevenness is compensated for by the APO scheme. Afterwards, the spectrum is pre-tilted again through the WSS and the second OA, and then sent to the subsequent fiber link. Finally, at the receiver side, the signal spectrum is, respectively, pre-amplified by two independent C band and L band EDFAs. The C band and L band signals are selected by corresponding WSSs and detected by the real-time coherent transponders. The bit error rates (BERs) and Q-factors can be recorded by the real-time transponders.

## 4. Results and Discussions

First, we investigate the required OSNR of the 400 Gbit/s signal at the C band and L band in the B2B case. The BERs before forward error correction (FEC) at different wavelengths in the C band and L band versus OSNR are shown in Figure 6. The results indicate that the required OSNRs at the C and L bands are almost the same. The required OSNRs are 17 dB at the FEC threshold (BER of $4 \times 10^{-2}$).

The optical power spectra before and after the APO scheme at the first OMS are recorded, as shown in the black and red lines of Figure 7, respectively. It can be observed that the maximum channel power of the high-frequency part of the C band is 9.4 dB less than the minimum channel power of the low-frequency part when APO is not imple-

mented, which is mainly caused by the SRS effect and results in bad performances at short wavelengths. Meanwhile, the low-frequency part of the L band has low power, which is different from the simulation, mainly caused by the uneven gain of the L band EDFAs used in the experiment. The L band OAs have lower gains in the lower frequency part, as shown in Figure 8. Moreover, the unevenness within the C band and L band is also due to the unevenness of the gain spectrum of the optical amplifier. After OMS power adjustment, the intra-band flatness of the C band and L band is clearly improved, from 9.4 dB to 2 dB for the C band and from 10 dB to 3 dB for the L band. In addition, the power flatness is modified from 12 dB to 4 dB over the entire 11 THz bandwidth, which shows the high efficiency of the OMS-based APO algorithm.

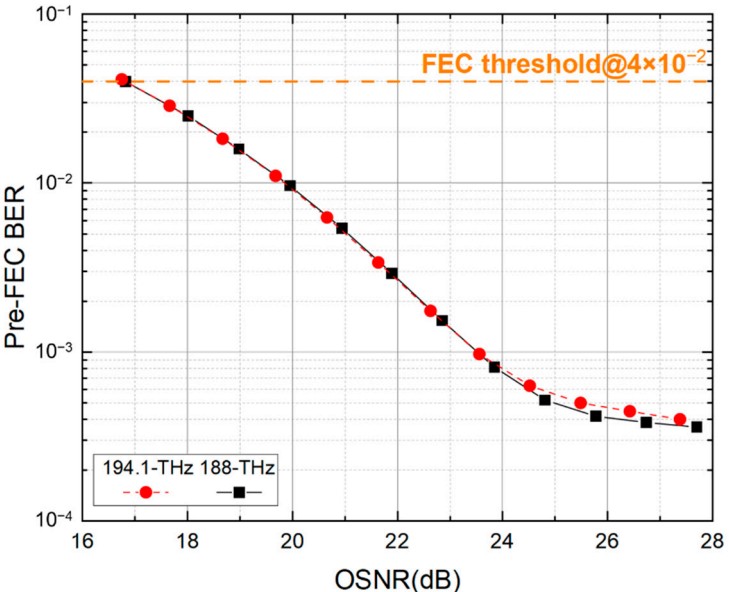

**Figure 6.** Pre-FEC BER performance versus OSNR in the C band and L band.

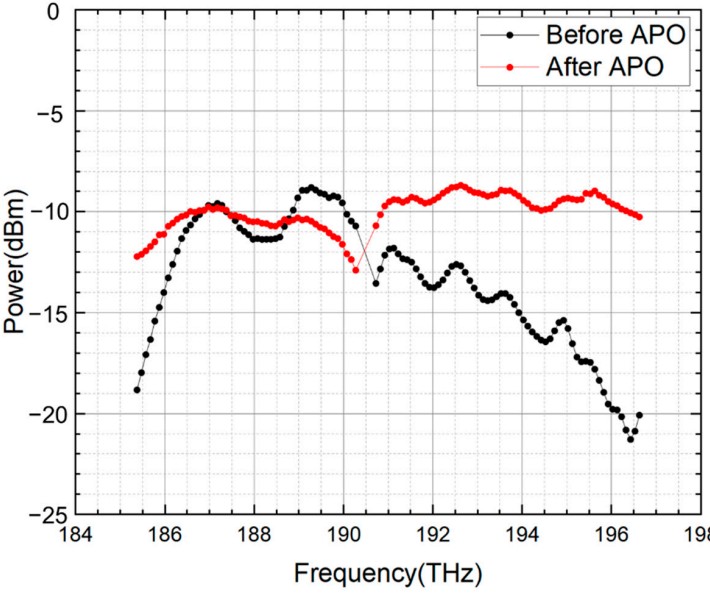

**Figure 7.** Optical power before and after the APO-scheme-based power adjustment.

Finally, the OSNR of all channels is measured across the widened C band and L band with APO, and the results are depicted in Figure 9. The received OSNR can be kept relatively flat (3.4 dB) across the 11 THz bandwidth, while the average OSNR of the C

band has 1.73 dB improvement compared to that of the L band. The long-term Q-factor performance for the C band and L band after the transmission is tested, as shown in Figure 10, by capturing the pre-FEC BERs at typical channels in both the C and L bands from the optical modules every 5 s and calculating the corresponding Q-factors. The results indicate that a very stable performance can be maintained over more than 1 h with almost less than 0.1 dB Q-factor fluctuations.

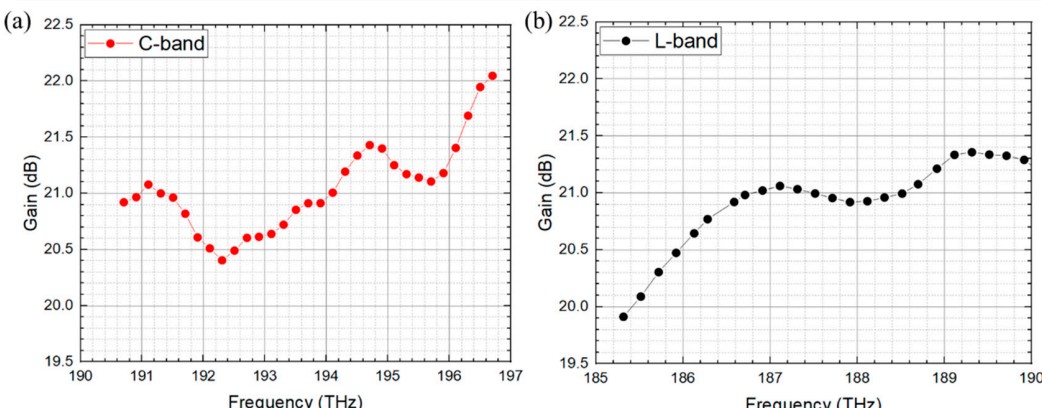

**Figure 8.** The gain spectrum of optical amplifiers for (**a**) the C band and (**b**) the L band.

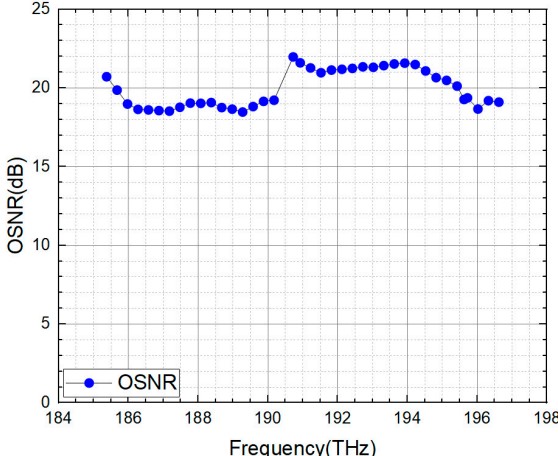

**Figure 9.** The OSNR performance versus frequency for the C band and L band after G.654.E fiber transmission.

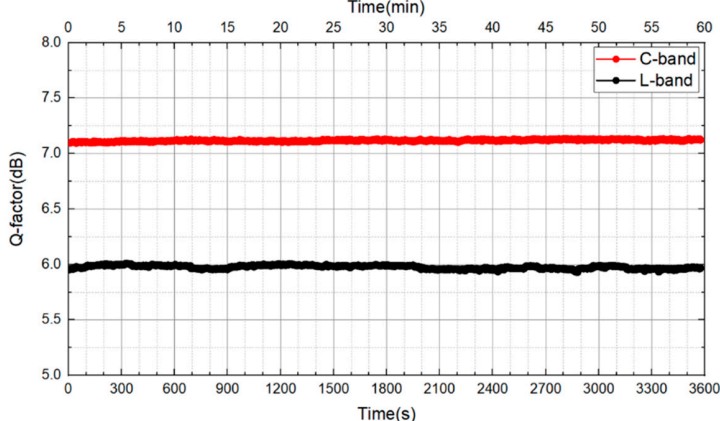

**Figure 10.** Long-term Q-factors for the C band and L band after G.654.E fiber transmission.

## 5. Conclusion

In this paper, we have numerically and experimentally investigated the impact of power unevenness, induced by the SRS effect, wavelength-dependent loss, and OA slope, in an 11 THz bandwidth system, and proposed the APO algorithm. Considering the cost of the actual system, the proposed APO algorithm has been successfully implemented in a simulation to achieve power flatness only by obtaining the power spectrum at the start and end of OMS in the entire fiber link. The results show that the APO algorithm reduces unevenness from 11 dB to 1.6 dB at the L band in the simulation. To demonstrate the feasibility of the proposed algorithm, we conduct a real-time 44 Tbit/s widened C+L transmission system over 1900 km G.654.E fiber utilizing 400 Gbit/s transponders. In the experiment, due to the gain spectrum of the optical amplifier itself, the low-frequency gain of the L band is relatively low, and the APO algorithm compensates for the power unevenness from 10 dB to 3 dB in the L band and from 9.4 dB to 2 dB in the C band. The received OSNR can be kept relatively flat (3.4 dB) across the 11 THz bandwidth. To the best of our knowledge, this is a record capacity and distance product (83.6 Pbit/s·km = 44 Tbit/s × 1900 km) in the real-time single-mode fiber transmission system.

**Author Contributions:** Conceptualization, A.Z. and X.H.; writing—original draft preparation, Y.L., L.F. and A.Z.; writing—review and editing, A.Z., L.F., K.L., H.L. and X.S.; experiment, A.Z., L.F., Y.L., H.C., Y.D. and J.W. All authors have read and agreed to the published version of the manuscript.

**Funding:** This research received no external funding.

**Institutional Review Board Statement:** Not applicable.

**Informed Consent Statement:** Not applicable.

**Data Availability Statement:** Data underlying the results presented in this paper are not publicly available at this time but maybe obtained from the authors upon reasonable request.

**Conflicts of Interest:** Authors Huan Chen and Yuting Du were employed by the ZTE Corporation company. Author Jun Wu was employed by the Yangtze Optical Fiber and Cable Joint Stock Limited Company. The remaining authors declare that the research was conducted in the absence of any commercial or financial relationships that could be construed as a potential conflict of interest.

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
