# Peer review of "Automatic Power Optimization of a 44 Tbit/s Real-Time Transmission System over 1900 km G.654.E Fiber and Widened C+L Erbium-Doped Fiber Amplifiers Utilizing 400 Gbit/s Transponders"

_photonics, doi:10.3390/photonics11010088_

Round 1

Reviewer 1 Report

Comments and Suggestions for Authors

The authors proposed an automatic power optimization algorithm to compensate the optical power slop in a DWDM system. The manuscript demonstrates promising simulation and results, which could contribute valuable insights.

11.  The OSNR after the algorithm was presented in the manuscript. As comparison, what’s the OSNR without the algorithm? Is the ONSR improved or degraded or a mix of both?

22. The Q-factor was also presented here. What’s the Q-factor without applying the algorithm?

33. Additionally, comparison of the pre-FEC BER with and without automatic power optimization algorithm would also strengthen the overall quality of the study.

Reviewer 2 Report

Comments and Suggestions for Authors

The manuscript “Automatic power optimization of a 44 Tbit/s real-time transmission system over 1900 km G.654. E fiber and widened C+L ED-FAs utilizing 400 Gbit/s transponders” proposes an automatic power optimization (APO) algorithm to guarantee reliable transmission for all channels. This algorithm effectively mitigates power unevenness by enhancing power flatness. To demonstrate the feasibility of the algorithm, the authors build a 5-span fiber system with 11 THz bandwidth in the simulation and experiments, and a recorded capacity and distance product is achieved in the real-time SMF transmission system. However, there are some elements that are still unclear and the manuscript lacks some significant research contents. In addition, my suggestions are as follows (see list below).

1. It is suggested to explain the abbreviations appearing in Figure 1 within the caption of Figure 1.

2. The authors should provide a detailed discussion of the impact of Stimulated Raman Scattering (SRS) on power unevenness and explain why it constitutes a significant challenge in multiband transmission systems.

3. Please provide a detailed explanation of how power flatness is calculated based on the power spectrum graph.

4. The author should carefully proofread the structure, syntax, and spelling of the article to ensure its coherence and accuracy. There are some spelling errors and expressions in the manuscript that need to be addressed.

On page 3, there is a semantic redundancy issue in the sentence: “Then, the first step (Step 1) is to compensate for the unevenness between the C band and L band by adjusting the gain of the C band and L band optical amplifiers to compensate for the power unevenness between the C band and L band.”

5. I would like to suggest that the paper explores the potential applications of relevant noise reduction techniques in the field of quantum communication, particularly in the context of long-distance quantum key distribution. It would be beneficial to cite and discuss relevant literature to highlight the broader implications and prospects of this work. Here, it is mandatory to include some important works on quantum key distribution.

[r1] Science Bulletin 67, 2167 (2022)

[r2] Nat. Photon. 17, 422–426 (2023)

[r3] Phys. Rev. Lett. 130, 250801 (2023)

[r4] Nat. Photon. 17, 416–421 (2023)

6. In the reference list, there is an issue with inconsistent author name formatting. While some references provide the names of all authors, others only list the first author's name followed by 'et al'. It is important to maintain a consistent author name format throughout the reference list to ensure uniformity and readability.

Comments on the Quality of English Language

Moderate editing of English language required

Round 2

Reviewer 2 Report

Comments and Suggestions for Authors

I recommend that this manuscript be accepted for publication.